# ArchCAD-400k: A Large-Scale CAD drawings Dataset and New Baseline for Panoptic Symbol Spotting

**Ruifeng Luo** [2,1*]**, Zhengjie Liu**[1,5*]**Tianxiao Cheng**[2]**, Jie Wang**[2]**, Tongjie Wang**[2]**, Xingguang Wei**[3,6]**, Haomin Wang**[3,7]**, YanPeng Li**[2]**, Fu Chai**[1,5]**, Fei Cheng** [1]**, Shenglong Ye** [3]**, Wenhai Wang**[3]**, Yanting Zhang**[8]**, Yu Qiao**[3]**, Hongjie Zhang**[3†]**, Xianzhong Zhao**[1,4†]

[1]Tongji University, [2]Arcplus East China Architectural Design & Research Institute Co., Ltd., [3]Shanghai AI Laboratory, [4]Shanghai Qi Zhi Institute [5]Shanghai Innovation Institute, [6]University of Science and Technology of China, [7]Shanghai Jiao Tong University, [8]Donghua University

`https://github.com/ArchiAI-LAB/ArchCAD`

## Abstract

Recognizing symbols in architectural CAD drawings is critical for various advanced engineering applications. In this paper, we propose a novel CAD data annotation engine that leverages intrinsic attributes from systematically archived CAD drawings to automatically generate high-quality annotations, thus significantly reducing manual labeling efforts. Utilizing this engine, we construct ArchCAD-400k, a large-scale CAD dataset consisting of 413,062 chunks from 5538 standardized drawings, making it over 26 times larger than the largest existing CAD dataset. ArchCAD-400k boasts an extended drawing diversity and broader categories, offering line-grained annotations. Furthermore, we present a new baseline model for panoptic symbol spotting, termed Dual-Pathway Symbol Spotter (DPSS). It incorporates an adaptive fusion module to enhance primitive features with complementary image features, achieving state-of-the-art performance and enhanced robustness. Extensive experiments validate the effectiveness of DPSS, demonstrating the value of ArchCAD-400k and its potential to drive innovation in architectural design and construction.

## 1 Introduction

For a long time, CAD drawings have served as the universal language of architectural design, enabling seamless communication among designers, engineers, and construction personnel through standardized graphical primitives such as arcs, circles, and polylines. Accurate perception of these primitives in 2D CAD drawings is essential for various downstream applications, including automated drawing review and 3D building information modeling (BIM) [1, 2, 3, 4], as this ability enables the optimization of CAD-based workflows while enhancing efficiency and precision in architectural design and construction processes[5, 6].

Building upon earlier research in symbol spotting [7, 8, 9, 10], pioneering work formally defined the task of panoptic symbol spotting [11] for floor plan CAD drawings and established the benchmark dataset FloorPlanCAD [11, 12, 13, 14, 15, 16]. However, the manual annotation of line-grained labels is a highly time-consuming and labor-intensive process, severely limiting the dataset's scale and diversity. This bottleneck restricts the ability to train models that can generalize across diverse

---

*Equal contribution;

†Corresponding Authors: nju.zhanghongjie@gmail.com; x.zhao@tongji.edu.cn.

39th Conference on Neural Information Processing Systems (NeurIPS 2025).

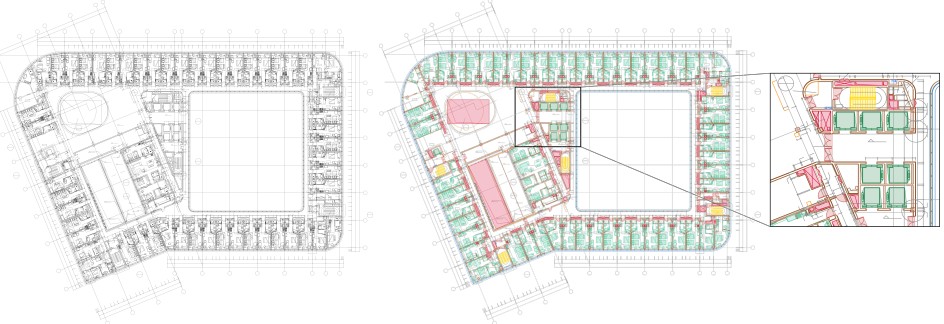

(a) Raw CAD Drawings          (b) Panoptic labeled Drawings

Figure 1: An example of annotated drawings in ArchCAD-400k. Each geometric primitive is assigned both semantic and instance labels, with distinct semantic categories represented by different colors and instances visualized through convex hull masks. The chunks are extracted from complete drawings.

building types, varying spatial scales, complex layouts, and a wide range of architectural component categories, thereby hindering their applicability to real-world scenarios.

To address these limitations, we propose an efficient annotation pipeline for line-grained labels, reducing annotation cost from 1,000 person-hours for 16K data to 800 hours for 413K. The core idea is to leverage the structured organization of floor plan drawings, including layers and blocks, to enable scalable and cost-effective large-scale annotation. Layers support bulk labeling via semantic grouping (e.g., doors, windows), while blocks allow instance reuse for repeated elements. As shown in Figure 2, the layer-block structure, inherently designed to enforce drawing standards and ensure consistency in architectural documentation, forms the foundational framework for our automated annotation pipeline. To ensure high data quality, we use completed drawings from top design institutions and a fully vectorized annotation workflow, with expert review of automated outputs.

Based on the efficient and accurate data engine, we construct ArchCAD-400k, a large-scale floor plan CAD drawing dataset for the panoptic symbol spotting task. The dataset contains 5,538 complete drawings meticulously selected from 11,917 industry-standard drawings, with a total of 413,062 annotated chunks, surpassing Floor-PlanCAD in scale by a factor of 26. While Floor-PlanCAD primarily focuses on residential buildings, ArchCAD-400k covers a diverse range of building types, with residential structures accounting for only 14% and large-scale public and commercial facilities forming the majority.

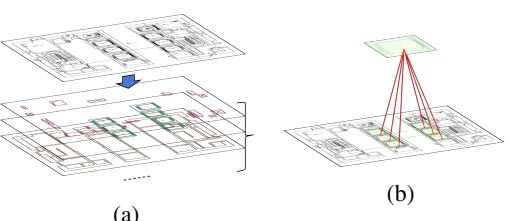

Figure 2: Layer-Block Structure: (a) Layers organize CAD drawings into functional categories. (b) Blocks define reusable elements for efficiency.

The average area of drawings in ArchCAD-400k spans $11{,}000\,\text{m}^2$, significantly larger than Floor-PlanCAD's $1{,}000\,\text{m}^2$ average, with 51.9% ranging from 1,000 to $10{,}000\,\text{m}^2$ and 4.4% exceeding $100{,}000\,\text{m}^2$. Furthermore, ArchCAD-400k introduces a comprehensive semantic categorization, including 27 categories such as structural components (*e.g.*, columns, beams), non-structural elements (*e.g.*, doors, windows), and drawing notations (*e.g.*, axis lines, labels), with 14 categories each containing over 1 million primitives. This extensive scale, diversity, and detailed annotation make ArchCAD-400k a robust resource for advancing AI models in construction industry. An example from ArchCAD-400k is illustrated in Figure 1.

We further propose a novel framework for panoptic symbol spotting, named Dual-Pathway Symbol Spotter (DPSS), which incorporates an adaptive fusion module to effectively enhance primitive features with complementary image features. This design enables the model to achieve superior performance and enhanced robustness compared to existing methods while demonstrating strong scalability and generalization capabilities on larger-scale datasets. In summary, our work presents the following contributions:

(1) We develop a highly efficient annotation pipeline specifically designed for floor plan CAD drawings, which generates high-quality annotations with improved efficiency compared to image-

based manual annotation, significantly reducing the annotation cost from 1,000 person-hours for 16K data to 800 person-hours for 413K data.

(2) We introduce ArchCAD-400k, a large-scale floor plan CAD drawing dataset that surpasses the current largest FloorPlanCAD dataset by an order of magnitude in size and exhibits greater diversity in terms of building types, spatial scales, and architectural component categories.

(3) We propose DPSS, a novel framework for panoptic symbol spotting that achieves state-of-the-art performance on FloorPlanCAD and ArchCAD-400k, surpassing the second-best method by 3% and 10%, respectively, with exceptional accuracy, robustness, and scalability.

## 2 Related Works

### 2.1 Floor Plan Datasets

Several datasets have been developed for floor plan analysis. SESYD [9] comprises 1,000 synthetic vectorized documents with ground truth annotations. FPLAN-POLY [10] contains 42 floor plans derived from images [17] for spatial relationship analysis. Cubicasa [3] provides 5,000 floor plan images annotated with over 80 object categories, focusing on residential layouts. RFP [18] includes 2,000 annotated floor plans with detailed room-level information for Asian residential buildings. FloorPlanCAD [11] offers 16,103 vector-graphic floor plans in ".svg" format, annotated across 35 object categories. LS-CAD [16] introduces a test set of 50 full-size CAD drawings with an average area of 1,000 square meters. However, these datasets are limited in scale and rely on labor-intensive annotation, hindering large-scale training.

### 2.2 Large-scale Vision Datasets

Large-scale vision datasets have significantly advanced deep learning models for complex visual understanding tasks[19, 20, 21, 22, 23, 24, 25, 26, 27], including floor plan recognition. Image-based datasets such as ImageNet [19] (14M images, 21K categories) and COCO [20] (300K images with object detection, segmentation, and captioning labels) have been instrumental in developing state-of-the-art models for object recognition, classification, and scene understanding. These datasets provide rich annotations that enhance model generalization, improving performance across general and domain-specific tasks. Nevertheless, large-scale vector-based datasets and efficient annotation processes for floor plan understanding remain scarce.

### 2.3 Panoptic Symbol Spotting

The panoptic symbol spotting task, initially proposed in [11], involves the simultaneous detection and classification of architectural symbols (*e.g.*, doors, windows, stairs) in floor plan CAD drawings. While traditional methods [7] focus on countable instances (*e.g.*, windows, tables), Fan *et al.* [11] extended this to uncountable objects (*e.g.*, walls, railings), inspired by [28]. They introduced PanCADNet, which integrates Faster R-CNN [29] for countable instances and Graph Convolutional Networks [30] for uncountable elements. Subsequently, Fan *et al.* [12] proposed CADTransformer, utilizing HRNetV2-W48 [31] and Vision Transformers [32] for primitive tokenization and embedding aggregation. Zheng *et al.* [15] adopted graph-based representations with Graph Attention Networks for semantic and instance-level predictions. Liu *et al.* [14] introduced SymPoint, exploring point set representations, later enhanced by SymPointV2 [13] through layer feature encoding and position-guided training. In contrast, CADSpotting [16] densely samples points along primitives and employs Sliding Window Aggregation for efficient panoptic segmentation of large-scale CAD drawings. While these methods perform well on FloorPlanCAD, they struggle on our more diverse, complex, and large-scale dataset, highlighting the need for robust and scalable solutions.

## 3 Annotation pipeline of ArchCAD-400k

We carefully gathered over 11917 complete CAD drawings from the industry, covering a wide range of building types. From the initial drawings, 5,538 drawings were validated as strictly conforming to the layer-block organizational standards. Through automated annotation combined with expert refinement, we generated approximately 413,062 chunks with line-grained annotations, each sized

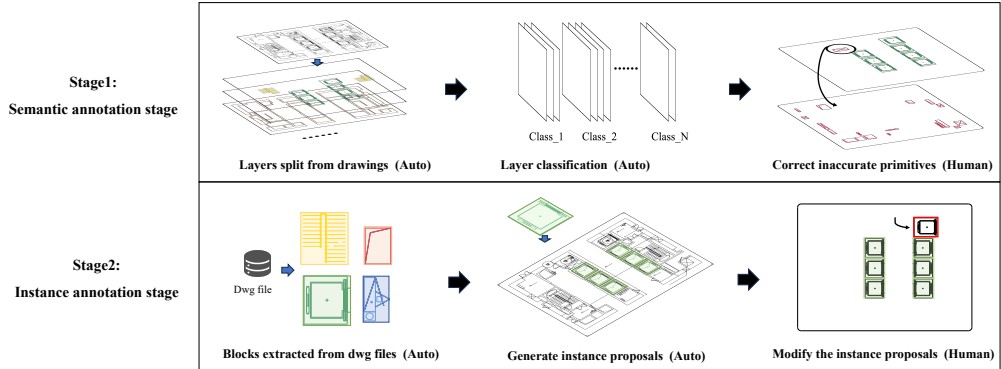

Figure 3: Overall pipeline of the annotation process. Primitives are labeled by automated methods followed by manual modification.

at 14m×14m, aligning with the design of FloorPlanCAD [11]. The overall annotation pipeline of ArchCAD-400k is shown in Figure 3 and detailed below.

## 3.1 Annotation Format

The ArchCAD-400k employs a structured annotation format tailored for the panoptic symbol spotting task, as defined in [11]. Each graphical primitive (e.g., line, arc, circle) in the drawing is characterized by a dual-identifier pair, $(l_k, z_k)$, where $l_k$ denotes its semantic category and $z_k$ represents the instance identifier. Primitives sharing the same $z_k$ value are considered to be part of the same instance.

## 3.2 Layer-Block Standardization Screening

Manual annotation of floor plans is inefficient due to crowded layouts and inconsistent symbols. Standardized drawings use layers for bulk annotation (e.g., door layer) and blocks for identifying repeated instances, clarifying topological and semantic relationships for automation.

However, the feasibility of automated annotation depends on the standardization level of the drawings. Non-standard drawings can have ambiguous layer names and mixed primitives, leading to semantic confusion. Notably, professional design institutes typically maintain internal layering standards, which can be normalized and processed to establish machine-interpretable structures. Thus, we restrict our data to completed drawings from leading design institutions and apply a layer naming validation algorithm against a reference table, discarding drawings with over 5% deviation. This dual quality control ensures standardization of the input data, providing a solid foundation for future automated annotation based on layer-block standardization. The reference table is provided in Appendix A.

## 3.3 Automated Annotation with Expert Refinement

The automated annotation process based on layer-block standardization greatly reduces manual workload and enables large-scale annotation. However, its accuracy is limited, especially with non-standard drawings that escape standardization. Semantic ambiguities and mixed primitives within layers often cause errors, necessitating human correction. To improve accuracy, we engaged 10 experienced architectural drafters with over 3 years of experience to refine annotations using a vector-based interface. Unlike typical image-based annotation methods such as bounding boxes or polygons, direct vector editing avoids raster-to-vector conversion errors and resolves issues like overlapping instances.

We adopted a two-stage annotation system as shown in Figure 3. In the semantic stage, layers are categorized by names and content, with their semantics displayed in for expert review and correction. In the instance stage, key categories are isolated and instance masks are proposed from block information, then manually refined. We also developed tools to automatically check layer-block compliance, flagging errors for quick human correction. Using this system, we annotated 5,538

Table 1: Comparison of ArchCAD-400k to existing CAD drawing datasets.

| Dataset | Source | Scale | | Data Format | | Annotation | | Elements | | |
|---|---|---|---|---|---|---|---|---|---|---|
| | | Size | Total Area | Raster | Vector | Type | Method | Architectural | Structural | Notation |
| FPLAN-POLY [10] | Internet | 48 | $<5 \times 10^4\,\mathrm{m}^2$ | ✓ | ✗ | Instance | Human | ✓ | ✗ | ✗ |
| SESYD [9] | Synthetic | 1000 | $<5 \times 10^5\,\mathrm{m}^2$ | ✓ | ✓ | Instance | Human | ✓ | ✗ | ✗ |
| FloorPlanCAD [11] | Industry | 16K | $1.6 \times 10^6\,\mathrm{m}^2$ | ✓ | ✓ | Panoptic | Human | ✓ | ✗ | ✗ |
| ArchCAD-400k | Industry | 413K | $8.1 \times 10^7\,\mathrm{m}^2$ | ✓ | ✓ | Panoptic | Auto & Human | ✓ | ✓ | ✓ |

drawings in 800 hours, which is over 10 times more efficient than the FloorPlanCAD dataset while maintaining high accuracy and reliability.

**Ethical and Copyright Considerations.** The dataset used in this work is derived from architectural or design drawings, which may contain potentially sensitive or proprietary information. To ensure compliance with ethical and copyright standards, we apply strict data anonymization before training or release, including removal of identifiable text and irreversible obfuscation of metadata, so that the original content of any individual project cannot be reconstructed. All data is used solely for research under academic fair use, and no raw data that may infringe copyright or IP rights is released. These measures ensure compliance with ethical standards and protection of PII.

# 4 Exploring ArchCAD-400k

We compare our ArchCAD-400k with existing datasets, as summarized in Table 1. Publicly available datasets, such as SESYD [9], FPLAN-POLY [10], and FloorPlanCAD [11], exhibit limitations in terms of source, scale, data format, annotation type and method, as well as the range of elements included in floor plans. These constraints affect their applicability to real-world architectural analysis.

## 4.1 Extended Building Diversity

Previous floor plan datasets predominantly originated from residential buildings, with a primary emphasis on indoor layouts. This limitation constrained their applicability in analyzing a wider variety of project types. In contrast, our dataset is curated from a more diverse array of buildings, as illustrated in Figure 4a, with residential structures comprising only 14% of the total. A substantial portion consists of large-scale public and commercial buildings, including office complexes, industrial parks, and other expansive facilities. This diverse composition provides a richer and more representative architectural dataset, enabling applications across a wider variety of building scenarios.

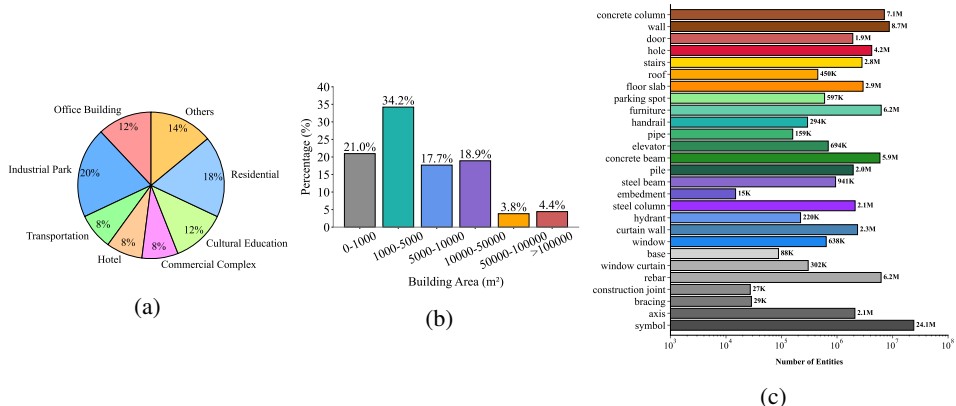

Figure 4: (a) Different project types of drawings; (b) Area distribution of drawings; (c) Number of annotated primitives for 27 semantic classes, where bar colors match each class's visualization.

## 4.2 Larger Data and Spatial Scale

ArchCAD-400k stands out for its unprecedented scale, comprising 5,538 complete drawings and 413,062 chunks, surpassing FloorPlanCAD's 15,663 chunks by over 26 times. This substantial scale significantly enhances the robustness and generalization capabilities of models for the spotting task.

In addition to the data scale, ArchCAD-400k features much larger drawings compared to existing datasets, which are generally restricted to small areas under $1,000\,\mathrm{m}^2$. The average drawing in our dataset spans $11,000\,\mathrm{m}^2$, closely reflecting real-world architectural dimensions. As shown in Figure 4b, 51.9% of the drawings cover areas between 1,000 and $10,000,\mathrm{m}^2$, while 4.4% exceed $100,000\,\mathrm{m}^2$, representing a wide spectrum of building areas and configurations.

## 4.3 Diverse Element Types

Traditional floor plan datasets typically focus on common non-structural elements, such as furniture, equipment, and decorative features. While such categorizations are sufficient for residential interior layouts, they often fail to capture critical components when applied to more diverse building types, particularly in complex architectural or industrial settings.

To address this limitation, our ArchCAD-400k adopts a comprehensive semantic categorization that extends beyond traditional architectural elements. Specifically, we introduce annotations for structural components (e.g., columns, beams, and holes) and drawing notations (e.g., axis lines, labels, and markers) in addition to non-structural elements. Structural components are fundamental to reconstructing the overall building structure as they define the load-bearing framework and spatial organization of the architectural design. In contrast, drawing notations are ubiquitous in real-world drawings and play a critical role in ensuring accurate interpretation and practical implementation. This tripartite categorization, encompassing non-structural, structural, and notation elements, enables a broader range of applications, including indoor navigation, architectural design, and structural analysis.

ArchCAD-400k provides detailed annotations for 27 categories, with the distribution of primitives across these categories visually represented in Figure 4c. Notably, 14 of these categories each encompass more than 1 million primitives, underscoring the extensive scale of our dataset. Additionally, our dataset includes 7 countable instance categories. While this count is fewer than that of FloorPlanCAD, our dataset surpasses it in terms of diversity and complexity within each category, presenting a more formidable challenge. For instance, we consolidate various fine-grained furniture types from FloorPlanCAD, such as chair, table, and bed, into a single unified "furniture" category. Visual examples of the categories can be found in Appendix A.

## 5 Method

To enhance panoptic symbol spotting on large-scale datasets, we propose **D**ual-**P**athway **S**ymbol **S**potter (DPSS), a robust framework free from prior knowledge like color or layer cues. As shown in Figure 5, DPSS employs a dual-path feature extractor: an image branch encodes rendered graphics for global context, while a point cloud branch captures geometric details by treating primitives as points. An adaptive fusion module aligns these multimodal features, and a transformer decoder performs panoptic segmentation for precise symbol spotting in complex diagrams.

### 5.1 Two-stream Primitive Encoding Module

The extraction of high-quality features for vector graphic panoramic segmentation is crucial as it directly affects segmentation performance. Unlike raster images, vector graphics consist of primitives, yet research on their modeling remains limited, highlighting the need for better feature extraction methods. CADTransformer [12] rasterizes vector graphics and samples features at the primitive centers, while SymPoint [14] encodes primitives as points using geometric attributes and a point cloud encoder[33]. Image-based approaches capture semantics well but struggle with fine-grained tasks, whereas point-based methods handle instances effectively but are less suited for sparse structures like walls. To address this, we combine visual and geometric encoding. Our visual encoder, based on HRNetV2 [34], extracts feature maps from rendered images. Meanwhile, a Point Transformer-

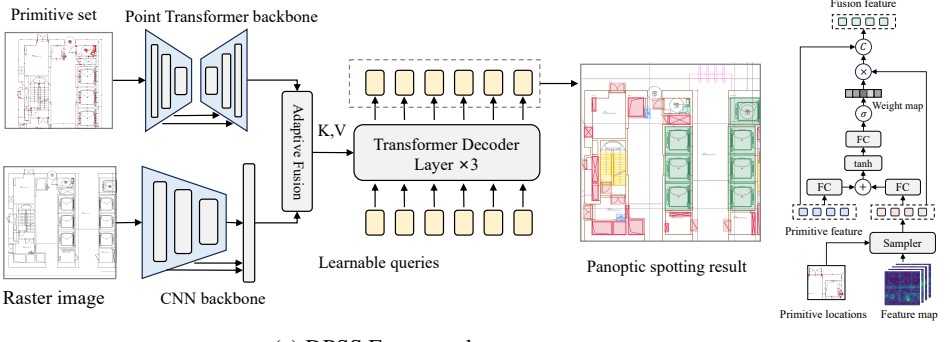

(a) DPSS Framework

(b) Adaptive fusion

Figure 5: The overall architecture of the proposed Dual-Pathway Symbol Spotter (DPSS). CAD drawings are encoded through both point backbone and image backbone. Features from dual pathways are aligned and fused in adaptive fusion module, then pass through a transformer decoder to generate mask predictions and classifications at primitive level.

inspired point cloud encoder generates geometric features for each primitive, ensuring a more comprehensive representation.

## 5.2 Adaptive Fusion Module

We use an image sampler to extract semantic features for each primitive. Given the primitive center $(x_c, y_c)$ and the feature map $F \in \mathbb{R}^{H \times W \times C}$, the sampler produces point-wise image features $V$. Since sampling points may fall between pixels, bilinear interpolation is applied:

$$V_p = \mathcal{K}(F_{N(p')}),\tag{1}$$

where $V_p$ is the extracted feature, $\mathcal{K}$ is the interpolation function, and $F_{N(p')}$ denotes neighboring pixels. The combination of primitive and image features is challenging due to size variations, noise, and overlapping. To address this, we introduce a geometry-guided fusion layer. First, graphic features $X_p$ and image features $V_p$ are transformed via an MLP, then concatenated and pass through an activation function $\sigma$. A weight matrix $w$ is computed as:

$$w = \sigma(W_1 \tanh(W_2 X_p + W_3 V_p)),\tag{2}$$

where $W_1, W_2, W_3$ are learnable parameters. Finally, the weighted features yield the representation:

$$U_p = \text{concat}(X_p, wV_p).\tag{3}$$

## 5.3 Decoder and Loss Function

The decoder is designed based on the architectures of DETR [35] and Mask2Former [36], achieving panoptic symbol spotting by predicting primitive-level masks along with their categories. The loss function is composed of a standard cross-entropy loss ($L_{cls}$) for class predictions, a binary cross-entropy loss ($L_{bce}$), and a Dice loss ($L_{dice}$) [37] for mask predictions. The overall loss is formulated as a weighted sum of the three losses $L = \lambda_{cls} L_{cls} + \lambda_{bce} L_{bce} + \lambda_{dice} L_{dice}$, where $\lambda_{cls}$, $\lambda_{bce}$, and $\lambda_{dice}$ denote the weight for each loss term respectively.

## 6 Experiments

We split our ArchCAD-400k dataset into training, validation, and test sets using a 7:1:2 ratio, ensuring that each drawing and its corresponding annotations appear in only one split. This results in 289,144 annotated samples for training, 41,306 for validation, and 82,612 for testing. Following the definition of panoptic symbol spotting, we evaluate the model performance using Panoptic Quality (PQ), Segmentation Quality (SQ), and Recognition Quality (RQ). The formulation of these metrics can be found in [11]. To evaluate model performance on ArchCAD-400k, we compare existing methods and our proposed method DPSS, with detailed results presented below.

Table 2: Panoptic symbol spotting results on FloorplanCAD [11] dataset

| Method | Additional prior inputs | Total | | | Thing | | | Stuff | | |
|---|---|---|---|---|---|---|---|---|---|---|
| | | PQ | SQ | RQ | PQ | SQ | RQ | PQ | SQ | RQ |
| SymPointV2[13] | w/ | 90.1 | 96.3 | 93.6 | 90.8 | 96.6 | 94.0 | 80.8 | 90.9 | 88.9 |
| CADSpotting[16] | w/ | 88.9 | 95.6 | 93.0 | 89.7 | 96.2 | 93.2 | 80.6 | 89.7 | 89.8 |
| DPSS | w/ | 89.5 | 96.2 | 93.1 | 90.4 | 96.6 | 93.5 | 79.7 | 91.1 | 87.5 |
| CADTransformer[12] | w/o | 68.9 | 88.3 | 73.3 | 78.5 | 94.0 | 83.5 | 58.6 | 81.9 | 71.5 |
| GAT-CADNet[15] | w/o | 73.7 | 91.4 | 80.7 | – | – | – | – | – | – |
| SymPoint[14] | w/o | 83.3 | 91.4 | 91.1 | 84.1 | 94.7 | 88.8 | 48.2 | 69.5 | 69.4 |
| SymPointV2[13] | w/o | 83.2 | 91.3 | 91.1 | 85.8 | 92.5 | 92.7 | 49.3 | 70.3 | 70.1 |
| DPSS | w/o | 86.2 | 93.0 | 92.6 | 88.0 | 94.1 | 93.5 | 64.7 | 83.0 | 77.9 |

## 6.1 Implement Details

We adopt HRNetW48 [31] pretrained on COCO-Stuff [38] as the backbone for the visual branch. The point cloud branch uses the PointTransformerV2 [39] encoder pretrained on ScanNetV2 [40]. The model is trained on 8 NVIDIA A800 GPUs and the optimizer is AdamW. On FloorplanCAD [11] dataset, we set a batch size of 2 per GPU and with a learning rate of $2 \times 10^{-4}$ and a weight decay of 0.1. The model is trained for 50 epochs. On ArchCAD-400k, we set a batch size of 4 per GPU and train the model for 10 epochs with a learning rate of $2 \times 10^{-4}$ and the same weight decay. Other baseline methods on ArchCAD-400k are trained for 10 epochs under their default configurations. All models were confirmed to have converged before evaluation to ensure fair performance comparison.

## 6.2 Quantitative Evaluation

**Quantitative comparison on FloorPlanCAD.** We evaluate multiple methods on the FloorPlanCAD dataset (Tables 2 and 3), under two scenarios: with and without prior information (such as layers and color). This distinction reflects how CAD drawings encode semantics—primitives in the same layer or with similar colors often share categories.

SymPointV2 improves on SymPointV1 by adding layer encoding, while CADSpotting samples primitives based on color and position. As shown in Table 2, DPSS performs comparably to these methods when priors are present. Without them, however, SymPointV2 suffers a notable drop, whereas DPSS stays robust—achieving 3% higher overall PQ and 15% higher "Stuff" PQ than SymPointV2, demonstrating better generalization. In Table 3, DPSS also clearly outperforms others in Semantic and Instance Spotting without priors, including some classical semantic segmentation and instance detection algorithms.

While our method benefits from prior information, we prioritize performance without it, as this better reflects real-world scenarios where such metadata is often unavailable.

Table 3: Semantic and instance spotting results on FloorPlanCAD

| Method | Additional prior inputs | Semantic spotting | | Instance spotting | | |
|---|---|---|---|---|---|---|
| | | F1 | wF1 | AP50 | AP75 | mAP |
| SymPointV2 [13] | w/ | 89.5 | 88.3 | 71.3 | 60.7 | 60.1 |
| CADSpotting [16] | w/ | 93.5 | 93.9 | 72.2 | 69.1 | 69.0 |
| DPSS | w/ | 93.1 | 93.2 | 74.8 | 71.0 | 70.8 |
| DeepLabv3+R101 [41] | w/o | 68.8 | 71.4 | – | – | – |
| DINO [42] | w/o | – | – | 64.9 | 54.9 | 47.5 |
| CADTransformer [11] | w/o | 82.2 | 80.1 | – | – | – |
| SymPoint [14] | w/o | 86.8 | 85.5 | 66.3 | 55.7 | 52.8 |
| SymPointV2 [13] | w/o | 87.0 | 86.3 | 66.4 | 57.7 | 57.5 |
| DPSS | w/o | 92.0 | 93.2 | 67.0 | 61.8 | 61.5 |

**Quantitative comparison on ArchCAD-400k.** We evaluate various methods on our ArchCAD-400k dataset (Table 4), comparing DPSS with CADTransformer, SymPoint, and SymPointV2, all without using layer or color priors, to better reflect real-world conditions. Compared to FloorPlanCAD, ArchCAD-400k is more challenging, leading to overall lower performance. For instance, SymPointV2 drops from 83.2% PQ on FloorPlanCAD to 60.5% on ArchCAD-400k —a decrease of over 22%.

Table 4: Panoptic, semantic, and instance symbol spotting results on ArchCAD-400k

| Method | Total | | | Thing | | | Stuff | | | Semantic Spotting | | Instance Spotting | | |
|---|---|---|---|---|---|---|---|---|---|---|---|---|---|---|
| | PQ | SQ | RQ | PQ | SQ | RQ | PQ | SQ | RQ | F1 | wF1 | AP50 | AP75 | mAP |
| CADTransformer[12] | 60.0 | 89.7 | 66.9 | 52.5 | 83.6 | 62.7 | 70.1 | 96.7 | 72.5 | 84.1 | 83.4 | — | — | — |
| SymPoint[14] | 47.6 | 86.1 | 55.3 | 51.4 | 91.9 | 55.9 | 39.9 | 73.9 | 54.0 | 76.8 | 62.0 | 36.1 | 30.9 | 30.8 |
| SymPointV2[13] | 60.5 | 88.0 | 68.8 | 62.4 | 91.7 | 68.1 | 52.8 | 73.7 | 71.7 | 69.8 | 69.3 | 44.9 | 40.2 | 39.7 |
| DPSS | 70.6 | 90.2 | 78.2 | 65.6 | 92.4 | 70.9 | 77.6 | 87.8 | 88.4 | 87.8 | 84.1 | 45.6 | 41.1 | 40.7 |

DPSS shows clear advantages in scalability and robustness. In semantic spotting, it achieves an F1 of 87.8%, outperforming CADTransformer (84.1%), SymPoint (76.8%), and SymPointV2 (69.8%). In instance spotting, it reaches 40.7% mAP, surpassing SymPointV2 (39.7%) and SymPoint (30.8%). For panoptic symbol spotting, DPSS attains a total PQ of 70.6%, significantly ahead of SymPointV2 (60.5%) and SymPoint (47.6%). Notably, its Stuff PQ reaches 77.6%, 24.8% higher than SymPointV2.

In summary, DPSS consistently outperforms baselines across all tasks, even without prior information. Its strong performance on the more complex ArchCAD-400k highlights its scalability, robustness, and practical applicability to real-world CAD scenarios.

## 6.3 Ablations

**Ablation studies on DPSS.** To demonstrate the effectiveness of the proposed DPSS, we conducted ablation studies focusing on different encoding strategies. Specifically, we evaluated the impact of using only the primitive encoder, only the image encoder, and the combination of both, as shown in Table 5 (Lines 1–3). The integration of both the image encoder and the primitive encoder yields a 1.3% improvement in PQ compared to using the primitive encoder alone, and a 2.1% gain over using only the image encoder. These results highlight the complementary nature of the two encoding branches. Furthermore, we investigated the role of the adaptive fusion module by replacing it with a simple concatenation strategy for combining image and primitive features, as shown in Table 5 (Lines 3–4). The results show that incorporating our adaptive fusion leads to a significant 3.2% increase in PQ, underscoring its effectiveness in enhancing feature integration.

Table 5: Ablation experiments on FloorPlanCAD

| Primitive Encoder | Image Encoder | Adaptive Fusion | PQ | RQ | SQ |
|---|---|---|---|---|---|
| ✓ | | | 81.7 | 90.1 | 90.6 |
| | ✓ | | 80.9 | 89.4 | 90.6 |
| ✓ | ✓ | | 83.0 | 90.1 | 92.1 |
| ✓ | ✓ | ✓ | 86.2 | 93.0 | 92.6 |

**Generalization ability of ArchCAD-400k.** To validate the generalization ability of ArchCAD-400k, We implement DPSS on FloorPlanCAD[11] dataset with a dual-pathway encoder pretrained on ArchCAD-400k. Results are illustrated in Figure 6a. The pre-trained model achieves faster and more stable convergence, surpassing the performance of non-pre-trained models at 50 epochs within 30 epochs. It suggests that ArchCAD-400k covers FloorPlanCAD and furthermore has more diverse patterns and properties for models to generalize across datasets.

**Scaling performance of ArchCAD-400k.** We conducted experiments to validate the scaling law in the context of panoptic symbol spotting. Using a subset of our ArchCAD-400k and adopting DPSS as the baseline network, we evaluated the training performance across different data scales, ranging from 10K samples to the full 400K samples. As demonstrated in Figure 6b and 6c, within 10K to 400K data range, doubling the dataset size consistently reduces the loss by 2.33 and improves the PQ metric by 6.40% in average. The results indicate that, for panoptic symbol spotting tasks, large-scale datasets significantly enhance model performance.

## 6.4 Qualitative Results

The qualitative results on our ArchCAD-400k are illustrated in Figure 7. In the presented cases, DPSS demonstrates superior performance than SymPointV2 [13]. In architectural diagrams (row 1), characterized by a diversity of component types and numerous interfering lines, DPSS is capable

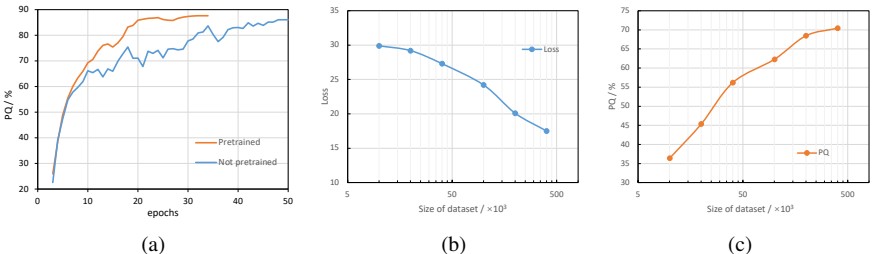

(a)        (b)        (c)

Figure 6: (a) Comparison of model performance: pretrained vs. non-pre-trained on ArchCAD-400k; (b) Loss convergence across different dataset sizes; (c) Panoptic quality across different dataset sizes.

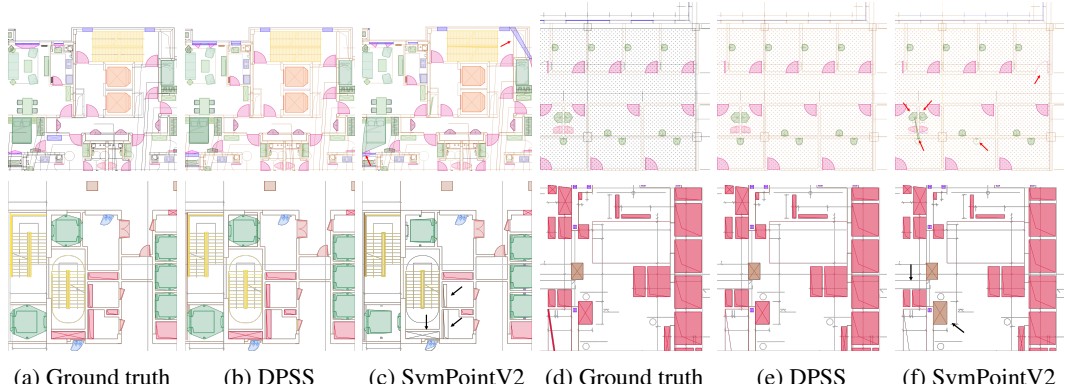

(a) Ground truth  (b) DPSS  (c) SymPointV2 (d) Ground truth  (e) DPSS  (f) SymPointV2

Figure 7: Qualitative comparison between DPSS and SymPointV2 [13] on FloorPlanCAD(Line1) and ArchCAD-400k (Line2).

of filtering out such disturbances to achieve precise spotting results. In structural drawings (row 2), where the shapes of components exhibit high similarity, DPSS effectively integrates contextual information to discern semantic differences between analogous components.

# 7 Conclusion

In this work, we address the challenges in panoptic symbol spotting for architectural CAD drawings by introducing an efficient annotation pipeline, a large-scale dataset (ArchCAD-400k), and a novel framework (DPSS). Our pipeline reduces annotation costs, enabling the creation of ArchCAD-400k, which surpasses existing datasets in scale and diversity. With 413,062 annotated chunks from 5,538 drawings, ArchCAD-400k covers diverse building types and spatial scales, advancing AI models in architectural design. DPSS, equipped with an adaptive fusion module to effectively enhance primitive features with complementary image features, achieves state-of-the-art performance on FloorPlanCAD and ArchCAD-400k, demonstrating superior accuracy, robustness, and scalability. We hope that our ArchCAD-400k will catalyze further progress in this domain.

# Acknowledgements

This work was supported by the Shanghai Qi Zhi Institute (SQZ202309), the National Natural Science Foundation of China (No. 62206046), and the Shanghai Committee of Science and Technology (No. 22YF1461500).

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

# A More Information About the ArchCAD-400k

## A.1 Detailed description of categorization

In Figure 8, we present some visual examples of content from each category. Engineering symbols from different categories can share great similarity. For example, simple rectangles can represent columns, holes, foundation, or furniture. Similarly, pairs of parallel lines might denote a pipeline, beam, or wall. At the same time, some instances can be drawn in diverse forms. As illustrated in Figure 8, there are at least six different ways to represent a door, and the appearance of stairs can vary greatly in different scenarios.

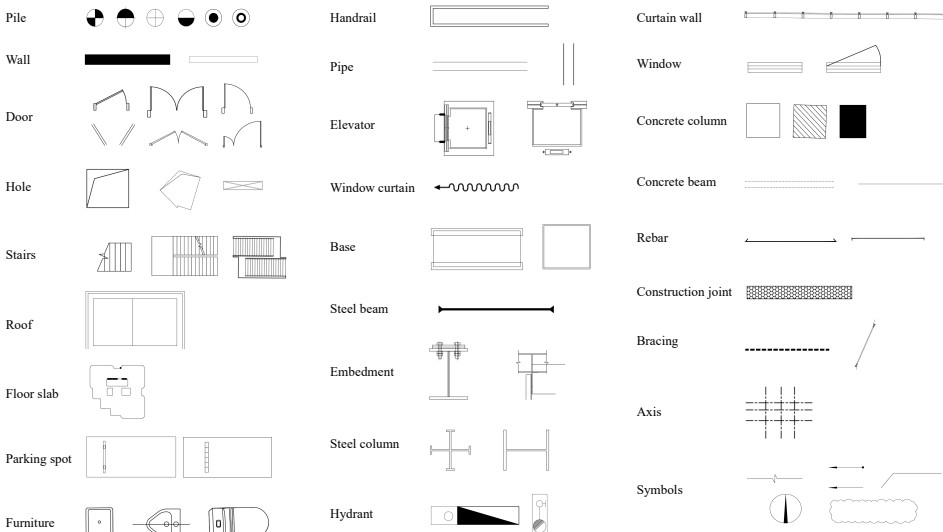

Figure 8: Visual examples of content in each category.

## A.2 Additional information about the annotation pipeline

Our annotation pipeline adopts regex matching to map layer names to semantic classes. The layer names in standardized drawings have a hierarchical format like [Discipline]-[Category]-[Modifier]. In Table 6, we list part of the standardized naming table for doors, walls, and stairs. Categories can be matched through the keyword in the table.

Table 6: Examples of correspondence between layer names and semantics

| Semantic label | Standard layer specifications | Content description |
|---|---|---|
| Door | A-DOOR | The surface structure of the door. |
| | A-DOOR-FRAM | The internal steel frame supporting the door. |
| | A-DOOR-HEAD | The line indicating the position of the door beam. |
| | A-DOOR-ROLL | The layout or design of the fireproof roller shutter. |
| | . . . | . . . |
| Wall | A-WALL-BLOK | Masonry block wall for structural stability. |
| | A-WALL-CONC | Concrete wall with high strength and fire resistance. |
| | A-WALL-STUD | Lightweight partition with stud framing and drywall. |
| | A-WALL-PRHT | Partial-height wall for spatial division. |
| | A-WALL-SCRN | Metal wall for electromagnetic shielding. |
| | A-WALL-FINI | Final surface treatment or cladding of a wall. |
| | A-WALL-INSU | Insulation layer for thermal or acoustic performance. |
| | A-WALL-TPTN | Partial-height wall for restroom privacy. |
| | A-WALL-EXPL | Reinforced wall to withstand blast pressures. |
| | S-WALL-LINE | The outline or framework of a wall in structural drawings. |
| | S-WALL-HATCH | Represents the filling or material pattern of a wall in structural drawings. |
| | . . . | . . . |
| Stairs | A-STRS-TREA | Stepping surface and vertical connector in stair construction. |
| | A-STRS-ESCL | Mechanical moving stairs for vertical transportation between floors. |
| | A-STRS-HRAL | Safety rails along stair edges for fall prevention. |
| | S-STRS-LINE | Outline representing stair geometry in structural drawings. |
| . . . | . . . | . . . |

## A.3 Additional examples of ArchCAD-400k

An example of well-labeled large architectural drawings exists in the main text. We further show two labeled drawings in our dataset in Figure 9. Each of them covers an area over $8000\,\mathrm{m}^2$, with various types of components.

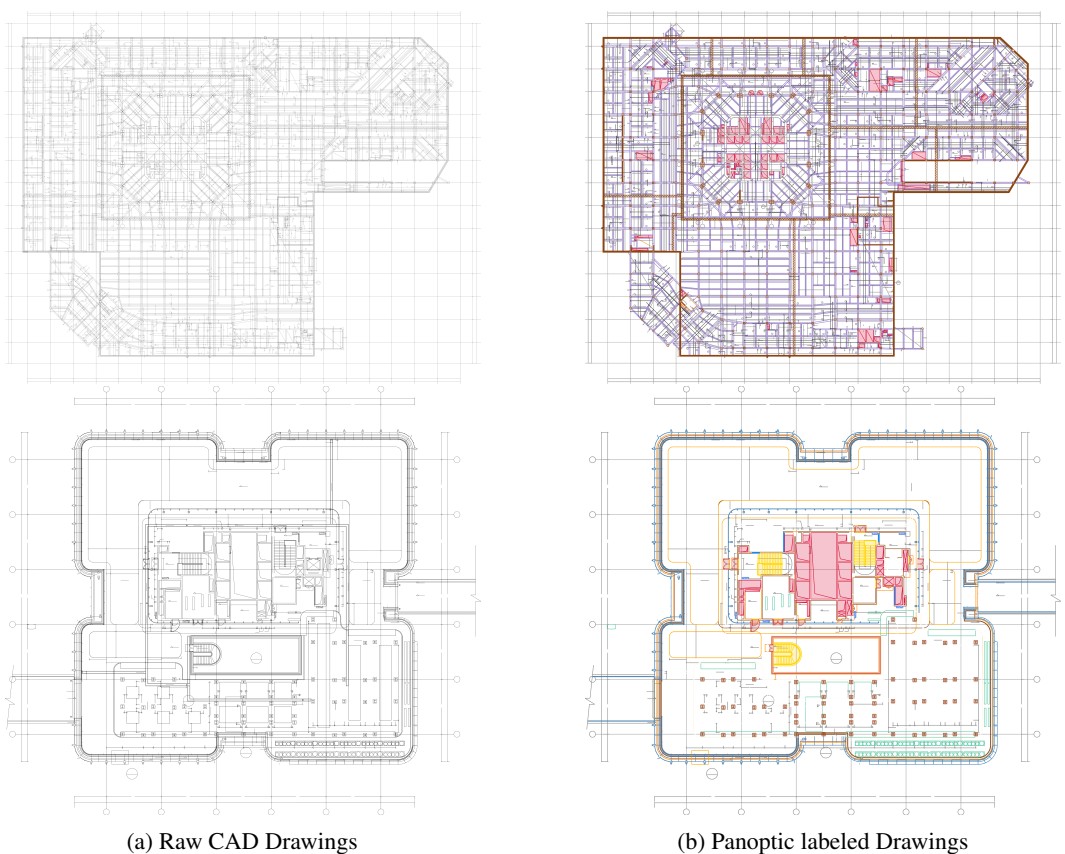

(a) Raw CAD Drawings        (b) Panoptic labeled Drawings

Figure 9: Additional example of the annotated drawings.

# B More Evaluations

## B.1 Analysis on Computational Efficiency

To evaluate the computational efficiency of the proposed method, we conducted additional experiments measuring inference latency across representative approaches. Our model operates on images with a resolution of 700×700, which introduces only a moderate computational overhead. Compared with SymPoint-V2 [13], the proposed method increases inference latency by approximately 30%, while achieving 3.0% and 10.1% higher panoptic quality on ArchCAD-400K and FloorPlanCAD, respectively. These results indicate that the proposed approach attains a favorable balance between computational efficiency and segmentation accuracy.

Table 7: Comparison with existing methods on FloorPlanCAD and ArchCAD-400K.

| Method | PQ (FloorPlanCAD) | PQ (ArchCAD-400K) | Average Latency |
|---|---|---|---|
| CADTransformer [12] | 68.9 | 60.0 | 159ms |
| SymPoint [14] | 83.3 | 47.6 | 66ms |
| SymPoint-V2 [13] | 83.2 | 60.5 | 95ms |
| DPSS (ours) | **86.2** | **70.6** | 121ms |

## B.2    Additional Quantitative Evaluations

We present the detailed experimental results of various methods on the ArchCAD-400k, including the panoptic quality (PQ) for each category as well as the mean IoU (Intersection over Union) for each category. The IoU for each category is obtained by calculating the intersection over union between the predicted panoptic segmentation masks and the ground truth masks. Different methods excel at handling different types of objects, DPSS achieves higher PQ and IoU metrics. For the current spotting results, there is still significant room for improvement.

Table 8: Quantitative results for panoptic symbol spotting of each category on ArchCAD-400k

| class | DPSS | | CADTransformer [11] | | SymPoint [14] | | SymPointV2 [13] | |
|---|---|---|---|---|---|---|---|---|
| | IoU | PQ | IoU | PQ | IoU | PQ | IoU | PQ |
| symbol | 84.4 | 89.8 | 62.4 | 70.7 | 76.4 | 80.9 | 74.2 | 79.3 |
| axis | 90.8 | 85.4 | 23.2 | 32.9 | 58.3 | 56.2 | 71.8 | 83.7 |
| door | 64.4 | 75.9 | 53.2 | 64.9 | 63.9 | 75.6 | 82.6 | 79.1 |
| floor slab | 40.8 | 85.6 | 14.4 | 80.3 | 20.9 | 82.8 | 47.7 | 69.4 |
| elevator | 45.6 | 77.6 | 26.5 | 63.9 | 54.6 | 73.3 | 28.0 | 69.2 |
| stairs | 60.4 | 65.4 | 35.3 | 45.2 | 41.3 | 55.4 | 43.7 | 78.0 |
| furniture | 72.5 | 88.7 | 66.5 | 73.3 | 63.2 | 83.9 | 43.4 | 54.0 |
| hole | 61.1 | 70.2 | 32.1 | 40.0 | 59.6 | 53.8 | 50.2 | 60.7 |
| window | 47.8 | 57.2 | 52.4 | 55.0 | 64.7 | 68.4 | 57.8 | 70.8 |
| curtain wall | 65.9 | 83.7 | 48.8 | 61.4 | 50.3 | 72.6 | 33.9 | 39.1 |
| wall | 73.4 | 75.6 | 39.1 | 45.9 | 52.7 | 57.7 | 42.3 | 70.7 |
| concrete column | 65.8 | 83.6 | 59.5 | 66.8 | 64.4 | 71.6 | 63.9 | 69.8 |
| steel column | 48.6 | 80.3 | 50.7 | 71.8 | 52.9 | 82.8 | 51.1 | 79.2 |
| concrete beam | 81.8 | 81.7 | 43.8 | 45.0 | 61.5 | 62.1 | 48.3 | 77.9 |
| steel beam | 79.1 | 73.0 | 19.7 | 34.9 | 47.5 | 52.2 | 71.5 | 77.4 |
| parking spot | 66.8 | 73.2 | 57.9 | 76.9 | 46.0 | 79.3 | 72.1 | 69.0 |
| roof | 3.5 | 28.7 | 3.7 | 39.1 | 5.7 | 50.1 | 55.6 | 76.1 |
| base | 85.0 | 90.3 | 65.8 | 75.2 | 71.2 | 83.1 | 16.3 | 21.3 |
| bracing | 19.6 | 33.8 | 7.8 | 20.0 | 20.2 | 23.2 | 67.6 | 81.1 |
| rebar | 72.7 | 93.1 | 39.4 | 69.4 | 56.5 | 81.1 | 53.5 | 75.9 |
| equipment | 37.2 | 45.8 | 1.0 | 7.0 | 30.8 | 31.8 | 28.4 | 29.1 |
| handrail | 73.6 | 65.9 | 31.2 | 31.0 | 40.2 | 42.7 | 70.3 | 85.8 |
| pipe | 44.4 | 48.3 | 32.9 | 34.4 | 51.7 | 41.7 | 52.5 | 42.4 |
| window curtain | 50.4 | 80.3 | 58.6 | 72.1 | 49.2 | 67.7 | 46.3 | 47.5 |
| construction joint | 0.0 | 5.3 | 0.0 | 0.3 | 2.5 | 4.8 | 4.1 | 3.7 |
| embedment | 0.0 | 0.0 | 0.0 | 0.0 | 0.0 | 0.0 | 1.2 | 2.8 |
| hydrant | 70.7 | 71.7 | 68.6 | 64.5 | 74.1 | 69.4 | 49.5 | 34.0 |
| overall | 70.6 | 67.04 | 47.6 | 49.1 | 60.5 | 59.1 | 60.0 | 60.8 |

## C    Limitations and Future Work

Although the proposed DPSS demonstrates strong performance on the panoptic symbol spotting task, some limitations remain. Notably, the current approach cannot process an entire vector drawing in a single pass, which leads to high computational overhead. With the availability of the ArchCAD-400k dataset, more complex and comprehensive research questions related to panoptic symbol spotting and the analysis of engineering line drawings can be explored. For example, future directions include the pre-trained models tailored for CAD vector graphics, and efficient inference strategies to handle large scale real-world drawings.

