# OpenReview forum: "ArchCAD-400K: A Large-Scale CAD drawings Dataset and New Baseline for Panoptic Symbol Spotting"
_NeurIPS.cc/2025/Conference — NeurIPS 2025 poster_

### Official Review · Reviewer_utxp · 2025-06-29

**Clarity:** 3
**Significance:** 3
**Originality:** 3
**Rating:** 5
**Confidence:** 1

**Summary:**

The authors introduce ArchCAD-400K, a large-scale CAD dataset with 413K annotated chunks from over 5,500 standardized drawings, created using a novel auto-annotation engine. They also propose DPSS, a new baseline for panoptic symbol spotting that fuses primitive and image features. DPSS achieves state-of-the-art performance, highlighting the dataset’s value for advancing CAD symbol recognition.

**Questions:**

Will the authors release the code, model, and dataset?

**Ethical Concerns:**

["NO or VERY MINOR ethics concerns only"]

**Final Justification:**

Thanks for addressing my concerns. So my rating is modified accordingly.

**Quality:**

3

**Strengths And Weaknesses:**

[Strengths]
* This paper is easy written.
* It is an important research problem that is worth exploration.
* The experimental results are good compared with the existing methods.

[Weaknesses]
* It is suggested to add the section to introduce the training time, the use of training resource, and inference time.
* The proposed framework is evaluated only on well-structured and curated datasets. There is no assessment of how the model performs under more challenging or noisy CAD environments, such as incomplete layers, overlapping primitives, or corrupted metadata, conditions common in practical settings. Could the authors explain more on it?
* The experimental results are reported without standard deviations, confidence intervals, or multiple runs. This omission makes it difficult to assess the reliability and statistical robustness of the claimed improvements.
* I did not see much weaknesses in this paper and I am not an expert in this domain.

---

> ### Author Rebuttal · Authors · 2025-07-30
>
> **Q1: About the release of the dataset and code.**
>
> **A1:** We fully intend to release all major components to the community.
>
> More specifically:
>
> (1) The dataset will be publicly released on HuggingFace before October 2025. We are currently undergoing final legal and technical checks to ensure proper anonymization and licensing.
>
> (2) In addition to the data, we will also release a toolkit designed for parsing, cleaning, and managing both our dataset and other CAD drawings from diverse sources. We believe this will significantly facilitate downstream research and application.
>
> (3) The code for the proposed DPSS framework will also be made available on GitHub along with the dataset toolkit.
>
> **Q2: About the reporting of training time, computational resources used, and inference time.**
>
> **A2:** Thank you for the valuable suggestion. We agree that reporting computational efficiency is important for a comprehensive understanding of the proposed method.
>
> We have included a summary of the training time and inference speed in the revised version as follows. Specifically, all experiments were conducted using 8 NVIDIA A800 GPUs. For the ArchCAD-400K dataset, we employed a batch size of 4 per GPU and trained the model for 10 epochs. For the FloorPlanCAD dataset, we used a batch size of 2 per GPU with 50 training epochs. Further details of the training and inference setup can be found in Section 6.1 of the main text.
>
> | Method | Training Time on ArchCAD-400K | Training Time on FloorPlanCAD | Average Inference Time |
> |:-----:|:------:|:------:|:------:|
> | CADTransformer[1] | 24h | 7h | 159ms |
> | SymPoint[2] | 40h | 18h |  66ms |
> | SymPoint-V2[3] | 48h | 13h | 95ms |
> | DPSS | 54h | 16h | 121ms |
>
> **Q3: About the framework’s performance on noisy or real-world CAD environments beyond curated datasets.**
>
> **A3:** We appreciate the reviewer's concern regarding data realism and robustness.
>
> First, regarding layer information, during the design of the panoptic symbol spotting task, we explicitly accounted for the possibility of incomplete or inconsistent layer annotations. Accordingly, our algorithmic evaluations were conducted under both with-layer and no-layer settings, as reflected in the "Additional prior inputs" column of Tables 2 and 3 in the main paper.
>
> Second, with respect to noise and imperfections in the drawings, our ArchCAD-400K dataset is collected directly from real-world architectural design archives, and thus naturally reflects the challenges and noise present in practical CAD environments.
>
> In summary, we have made a deliberate effort to ensure that both our dataset and experimental setup reflect realistic conditions encountered in actual CAD workflows.
>
> **Q4: About reporting the standard deviations and multiple experimental runs.**
>
> **A4:** We acknowledge this limitation. Due to the considerable computational cost associated with training (as noted in Q2), it is difficult to perform multiple full runs for each setting. We plan to include repeated experiments for key configurations in the revised version to further validate the stability of our results.
>
> Nevertheless, to partially mitigate this, we used consistent data splits and evaluation protocols across all methods to ensure fair comparison. Besides, the performance gaps between our method and baselines are obvious across both datasets (Table 2, 3, 4), which we believe provides sufficient support for the reliability of our claims.
>
> **References**
>
> [1] Zhiwen Fan, Tianlong Chen, Peihao Wang, and Zhangyang Wang. Cadtransformer: Panoptic symbol spotting transformer for cad drawings. In Proceedings of the IEEE/CVF Conference on Computer Vision and Pattern Recognition, pages 10986–10996, 2022.
>
> [2] Wenlong Liu, Tianyu Yang, Yuhan Wang, Qizhi Yu, and Lei Zhang. Symbol as points: Panoptic symbol spotting via point-based representation. arXiv preprint arXiv:2401.10556, 2024.
>
> [3] Wenlong Liu, Tianyu Yang, Qizhi Yu, and Lei Zhang. Sympoint revolutionized: Boosting panoptic symbol spotting with layer feature enhancement. arXiv preprint arXiv:2407.01928, 2024.

---

> > ### Comment · Reviewer_utxp · 2025-08-09
> > **Feedback about rebuttal**
> >
> > The rebuttal has addressed all of my concerns. I have also checked the discussions between other reviewers have felt satisfied about it. So I increase the rating accordingly. Hope the authors will revise the paper for the final version.

---

### Official Review · Reviewer_NcKG · 2025-07-02

**Clarity:** 3
**Significance:** 3
**Originality:** 3
**Rating:** 4
**Confidence:** 5

**Summary:**

This paper proposes a large-scale floor plan CAD drawing dataset called ArchCAD-400k, which contains 413K CAD chunks from 5,538 full drawings. In addition, the authors introduce a Dual-Pathway Symbol Spotter for CAD spotting, which simultaneously leverages rendered CAD image features (extracted by HRNetV2) and symbol primitive features (extracted by SymPoint) to perform symbol spotting in CAD drawings. Experimental results demonstrate that the proposed dataset improves model performance.

**Questions:**

One major contribution of this paper is providing a large-scale CAD dataset. Will this dataset be released in the future? If so, what is the release timeline?

**Ethical Concerns:**

["NO or VERY MINOR ethics concerns only"]

**Final Justification:**

I have carefully read the authors' response and feel that they have addressed most of my concerns. The only remaining issue is that, although adding an image feature extraction module significantly improves performance, the novelty is somewhat limited. However, considering that the authors will release the proposed dataset, which is valuable to the community, I will maintain my original rating.

**Limitations:**

The authors did not discuss limitations in the main text. In my opinion, the biggest limitation of the proposed method is that the image-based feature extraction becomes time-consuming when processing high-resolution CAD drawings, especially compared to point-based methods such as SymPoint or SymPointv2.

**Quality:**

3

**Strengths And Weaknesses:**

### Strengths
1. The paper introduces a large-scale floor plan CAD drawing dataset, ArchCAD-400k, and demonstrates an efficient annotation pipeline.
2. It proposes a Dual-Pathway Symbol Spotter method for CAD symbol spotting, achieving state-of-the-art results.
### Weaknesses
1. Combining the advantages of image-based methods (CADTransformer) and point-based methods (SymPoint) and extracting features through a dual-path approach is reasonable, but the novelty is moderate and this design increases computational cost.
It would be better if the authors could compare the computational speed of their proposed method with other SOTA methods such as CADTransformer and SymPoint. For high-resolution images, feature extraction is expected to be quite time-consuming.
2. The design of the decoder is the same as SymPoint/SymPointv2, so proper citation should be added. The loss function involves multiple weights for balancing; it would also be important to include ablation studies showing the impact of these weights on the results and convergence.

---

> ### Author Rebuttal · Authors · 2025-07-30
>
> **Q1: About computational speed of the proposed method.**
>
> **A1:** We appreciate the reviewer's suggestion regarding the comparison of computational speed with other SOTA methods.
>
> Our model operates on images with a resolution of 700×700 pixels, which introduces some additional computational cost, but the increase is modest due to the relatively small size. We selected this resolution empirically—higher resolutions did not yield further performance gains, while lower resolutions led to noticeable performance degradation.
>
> We also found that incorporating image features significantly improves model performance. These features enhance the robustness to noisy primitives and contribute positively to overall panoptic quality, making the moderate computational overhead worthwhile.
>
> We measured the average inference latency of several representative methods on an NVIDIA A800 GPU, as summarized in the table below. Compared to SymPoint-V2[3], our method increases latency by only 30%, while achieving a 3.0% and 10.1% improvement in panoptic quality on ArchCAD-400K and FloorPlanCAD respectively.
>
> | Method | PQ (FloorPlanCAD) | PQ (ArchCAD-400K) | Average Latency |
> |:------:|:--------:|:--------:|:------:|
> | CADTransformer[1] | 68.9 | 60.0 | 159ms |
> | SymPoint[2] | 83.3 | 47.6 | 66ms |
> | SymPoint-V2[3] | 83.2 | 60.5 | 95ms |
> | DPSS | 86.2 | 70.6 | 121ms |
>
> **Q2: About the impact of loss function weighting.**
>
> **A2:** We conduct some experiments to determine a reasonable configuration of loss function. Specifically, we experimented with different combinations of loss weights $\lambda_{cls}$ : $\lambda_{bce}$ : $\lambda_{dice}$, and empirically identified a reasonable range for balancing these terms. The setting around 1:2:2 yielded stable performance and was adopted as the default in our implementation.
>
> | $\lambda_{cls}$ | $\lambda_{bce}$ | $\lambda_{dice}$ | PQ | RQ | SQ |
> |:------:|:------:|:------:|:------:|:------:|:------:|
> | 1 | 0.5 | 0.5 | 82.4 | 89.5 | 92.1 |
> | 1 | 1 | 1 | 84.7 | 92.7 | 91.4 |
> | 1 | 2 | 2 | 86.2 | 93.0 | 92.6 |
>
> **Q3: About the release of the dataset.**
>
> **A3:** We fully intend to release all major components to the community.
>
> More specifically:
>
> (1) The dataset will be publicly released on HuggingFace before October 2025. We are currently undergoing final legal and technical checks to ensure proper anonymization and licensing.
>
> (2) In addition to the data, we will also release a toolkit designed for parsing, cleaning, and managing both our dataset and other CAD drawings from diverse sources. We believe this will significantly facilitate downstream research and application.
>
> (3) The code for the proposed DPSS framework will also be made available on GitHub along with the dataset toolkit.
>
> **References**
>
> [1] Zhiwen Fan, Tianlong Chen, Peihao Wang, and Zhangyang Wang. Cadtransformer: Panoptic symbol spotting transformer for cad drawings. In Proceedings of the IEEE/CVF Conference on Computer Vision and Pattern Recognition, pages 10986–10996, 2022.
>
> [2] Wenlong Liu, Tianyu Yang, Yuhan Wang, Qizhi Yu, and Lei Zhang. Symbol as points: Panoptic symbol spotting via point-based representation. arXiv preprint arXiv:2401.10556, 2024.
>
> [3] Wenlong Liu, Tianyu Yang, Qizhi Yu, and Lei Zhang. Sympoint revolutionized: Boosting panoptic symbol spotting with layer feature enhancement. arXiv preprint arXiv:2407.01928, 2024.

---

> > ### Comment · Reviewer_NcKG · 2025-08-05
> >
> > Thank you for the authors' response. They have carefully addressed my concern about the computational cost of the newly added image feature extraction part. It seems that the computational cost is modest and the performance is substantially improved. The authors have given a clear timeline for the release of the dataset, which addresses one of my other concerns. Therefore, I will keep my initial rating.

---

### Official Review · Reviewer_FDkf · 2025-07-02

**Clarity:** 2
**Significance:** 3
**Originality:** 3
**Rating:** 5
**Confidence:** 3

**Summary:**

This paper presents a dataset and learning algorithm for panoptic symbol spotting: finding and categorizing symbols in 2D CAD drawings. The dataset is much larger in terms of total floorplan area than previous datasets. It was created semi-automatically by automatically proposing segmentations and labeling to expert reviewers for refinement (this process is also a contribution). The proposed symbol spotting architecture uses parallel CNN and Point Transformer backbones to jointly encode raster and primitive input that are fused before being decoded by a transformer decoder into the final panoptic spotting results. The proposed spotting algorithm outperforms previous methods over both the contributed dataset and the previously largest dataset.

**Questions:**

Figure 4(a): Is this a breakdown by chunks or by source drawings? If it is by drawings, what does the breakdown look like by chunks? I am concerned about bias because the larger industrial buildings presumably make up a far larger percentage of chunks than smaller residential plans due to having larger floorplan areas.

Please explain how the initial proposals in the expert labeling process were generated. Is it done entirely from authored layer annotations, or are there other steps (for examples data-driven predictive methods) involved?

What are the "Thing" and "stuff" metrics? These are never defined in the exposition.

**Ethical Concerns:**

["Major Concern: Data privacy, copyright, and consent"]

**Final Justification:**

I maintain my score of accept. In the rebuttal, the reviewers addressed my primary concerns about dataset bias and reproducibility of their labeling methodology, as well as promising to add more qualitative results to give context to the quantitative. If these and the clarification on the dataset size comparison to FloorPlanCAD are included in their final revision, I am happy to accept this paper.

**Limitations:**

I am unclear if the claim about no IP being shared on line 157 is true (e.g. do the floorplan's themselves constitute IP even with metadata and identifiable labels scrubbed?) The only note about how the raw data was collected is on 113: "We carefully gathered ... complete CAD drawings from the industry," so it is difficult to evaluate the claims of 152-158 (and just looking at the example data from the supplemental, it seems likely that the individual chunks could be reassembled to reconstruct full floorplans).

**Quality:**

3

**Strengths And Weaknesses:**

Strengths:
- The dataset collected is very large _and_ expertly curated, and because the curation strategy is documented, it should be possible for anyone to expand that dataset using the same methodology.
- Is SOTA in the more flexible no-priors case, so should generalize better than existing methods to less clean data.


Weaknesses:
- The dataset proposed in this work reduces the granularity of categorization in some areas compared to previous datasets (e.g. collapsing many categories down into "furniture"
- This dataset's claims to be the largest by an order of magnitude is true on the measure of "chunks", which is directly related to the total floorplan area covered, it is actually built off of fewer overall distinct drawings than FloorPlanCAD. This apparently comes from the inclusion of larger public and commercial buildings in the dataset, which may bias the dataset, and ultimately provide less diversity than is implied by the dataset's size
- It is unclear how the initial proposal annotations given to experts for refinement are generated
- Few qualitative results shown, and without connection to quantitative results to aid in qualitative understanding of what the quantitative values mean.

---

> ### Author Rebuttal · Authors · 2025-07-30
>
> **Q1: About the reduction in categorization granularity compared to previous datasets, such as merging multiple categories into "furniture".**
>
> **A1:** Thank you for the observation. We intentionally chose to reduce the categorization granularity for certain object types. For example, we grouped subtypes such as tables, chairs, and cabinets into a single furniture category. Our rationale is twofold:
>
> (1) Limited Utility: While structural elements such as walls, columns, and curtain walls are essential for architectural modeling and reconstruction, fine-grained distinctions among furniture types offer relatively limited benefit for these purposes. The impact of individual furniture subtypes on the global structure is minimal, making detailed categorization a lower priority.
>
> (2) Annotation Efficiency: Preliminary analysis revealed that the furniture class could be subdivided into over 20 distinct subtypes. However, annotating such fine-grained labels across large-scale datasets would introduce substantial overhead. Given the limited structural value of these distinctions, we opted for a more efficient, coarser labeling strategy.
>
> Overall, this reflects a conscious trade-off between annotation cost and semantic value, aligned with the objectives of our reconstruction task.
>
> **Q2: About the relationship between dataset size, the number of distinct drawings, and potential diversity bias due to the inclusion of larger buildings.**
>
> **A2:** We appreciate this important point. Our dataset contains a larger number of distinct drawings than FloorPlanCAD. FloorPlanCAD has 1,332 distinct CAD drawings. In contrast, our dataset includes 5,538 distinct drawings.
>
> Both datasets are organized in a chunk-wise fashion, but the increased number of large buildings in our dataset reflects an expansion in both spatial coverage and complexity, rather than a reduction in diversity. We acknowledge the concern that larger buildings could skew the distribution; however, we take steps to mitigate this, as described in our response to Q3.
>
> **Q3: About the basis of distribution in Figure 4(a), and the potential bias introduced by chunk-wise representation.**
>
> **A3:** Thank you for raising this concern. Figure 4(a) indeed shows the distribution by source drawings, not by chunks. We recognize that the correlation between drawing type and spatial size could result in an overrepresentation of certain sources at the chunk level.
>
> To address this, we conducted an additional analysis of the dataset with a chunk-wise breakdown, presented in the as follows.
>
> | Category | By Drawings | By Chunk |
> |:---------------|:------:|:------:|
> | Office Building  | 12% | 14% |
> | Industrial Park | 20% | 21% |
> | Transportation | 8% | 17% |
> | Hotel | 8% | 12% |
> | Commercial Complex | 8% | 13% |
> | Cultural Education | 12% | 6% |
> | Residential | 18% | 12% |
> | Others | 14% | 5% |
>
> This analysis revealed that while there are some differences in proportions between drawing-wise and chunk-wise distributions, the semantic category distribution and spatial diversity remain well-balanced at the chunk level.
>
> **Q4: About the generation of initial annotation proposals.**
>
> **A4:** We appreciate the interest in our annotation workflow. The initial labeling proposals were generated entirely through CAD layer-based heuristics, without any machine learning or predictive components.
>
> Specifically, we built a layer-to-category mapping table by consolidating naming conventions across various CAD standards and companies. This mapping enabled automatic pre-labeling of most graphical elements. Annotators then reviewed and refined these proposals based on context and visual correctness.
>
> A partial snapshot of this mapping table is provided in Appendix A.2. This rule-based pipeline ensured high initial labeling quality and interpretability, which was subsequently verified through multi-stage quality control.
>
> **Q5: About the qualitative results.**
>
> **A5:** We appreciate the reviewer's suggestion. In the revised version, we will include additional qualitative visualizations to better illustrate the strengths and limitations of our method. These visual examples will be carefully selected to correspond with the quantitative metrics, thereby providing more intuitive insights into what the reported numerical values reflect in practice.
>
> **Q6: About the definition and usage of the terms "Thing" and "Stuff" in evaluation metrics.**
>
> **A6:** Thank you for highlighting this point. The terms “Thing” and “Stuff” come from definition of panoptic symbol spotting [1].
>
> "Things" refer to countable, instance-level objects (e.g., columns, doors).
>
> "Stuff" refers to amorphous symbols of similar objects or material with semantic meaning (e.g., wall).
>
> Evaluating the two metrics separately allows for a more precise assessment of the model's performance on semantic and instance segmentation tasks, which is a common practice on this problem [1–3].
>
> **Q7: About the handling of intellectual property in the dataset.**
>
> **A7:** We acknowledge your concern and agree that raw floorplans could constitute IP, even if scrubbed of metadata and labels. During the data processing period, we applied several measures for IP protection:
>
> (1) Spatial Chunking: Floorplans are divided into localized chunks, containing only partial spatial context.
>
> (2) Cross-Source Shuffling: Chunks are decoupled from their original drawing context and mixed across different sources.
>
> (3) Text Anonymization: Any textual elements that could reveal project background (like project names, client identifiers, etc.) are removed.
>
> Due to this process, it is extremely difficult to reconstruct the original full floorplan, even with access to all chunks. Some of the shuffled chunks are shown in supplementary materials. We believe this approach significantly reduces the risk of re-identification or IP leakage.
>
> **References**
>
> [1] Zhiwen Fan, Lingjie Zhu, Honghua Li, Xiaohao Chen, Siyu Zhu, and Ping Tan. Floorplancad: A large-scale cad drawing dataset for panoptic symbol spotting. In Proceedings of the IEEE/CVF international conference on computer vision, pages 10128–10137, 2021.
>
> [2] Zhiwen Fan, Tianlong Chen, Peihao Wang, and Zhangyang Wang. Cadtransformer: Panoptic symbol spotting transformer for cad drawings. In Proceedings of the IEEE/CVF Conference on Computer Vision and Pattern Recognition, pages 10986–10996, 2022.
>
> [3] Wenlong Liu, Tianyu Yang, Yuhan Wang, Qizhi Yu, and Lei Zhang. Symbol as points: Panoptic symbol spotting via point-based representation. arXiv preprint arXiv:2401.10556, 2024.

---

> > ### Comment · Reviewer_FDkf · 2025-08-04
> >
> > Thank you for your detailed responses, and for running the additional chunk-level evaluation!
> >
> > Q2: Line 84 of your paper states that FloorPlanCAD has 16,103 floor plans; how does this number relate to the 1,332 claimed in the rebuttal (is the 16k equivalent to your chunk count, and if so, are they scaled similarly?).
> >
> > Q4: Thank you for pointing out the mapping in A.2. Please make a full version of this available somewhere for reproducibility of the method.
> >
> > Q7: Thanks for clarifying the term of sharing with the ethics reviewers. I'm not convinced that cross-source shuffling makes reconstruction particular difficult for someone with access to the full dataset, but it sounds like the IP of the floorplans for this particular dataset is not a privacy concern anyway due to the terms of licensing?

---

> > > ### Author Response · Authors · 2025-08-05
> > >
> > > **A2:** Thank you for raising this question. The misalignment comes from two different ways of quantifying dataset size.
> > >
> > > As stated in the FloorPlanCAD paper, the number 16K refers to the total number of spatial chunks, each covering a 10m × 10m area. In contrast, our dataset contains 413K chunks of 14m × 14m size. These chunk-based statistics reflect the actual training units used in models.
> > >
> > > In the rebuttal, we referred to a different statistic, the number of complete drawings. Our dataset includes 5,538 complete drawings, while FloorPlanCAD contains 1,332 complete drawings, as verified through inspection of their dataset. This drawing based count offers a complementary perspective on dataset scale and diversity.
> > >
> > > (Note: While the FloorPlanCAD paper does not explicitly report the number of complete drawings, their dataset includes clear source drawing identifiers. It is straightforward to verify that the number of drawings is 1,332.)
> > >
> > > **A4:** As mentioned in our open-source plan (see discussion with reviewer NcKG, Q3), in addition to releasing the dataset, we will provide a comprehensive toolkit designed for parsing, cleaning, and managing both our dataset and other CAD drawings.
> > >
> > > The complete mapping described in Appendix A.2 is included as part of this toolkit, enabling users to interpret CAD drawings from other diverse sources.
> > >
> > >
> > > **A7:** Thank you for the follow-up.
> > >
> > > We acknowledge that a determined actor could theoretically attempt to reconstruct parts of the original floorplans despite our cross-source shuffling strategy. However, the shuffling strategy significantly reduces the likelihood and ease of such reconstruction, especially at scale.
> > >
> > > More importantly, as discussed in our response to the ethics reviewers, the raw CAD files were obtained through formal research collaboration agreements with top-tier architectural design institutions. These agreements authorize the use of the data for research purposes and address intellectual property and privacy concerns. The dataset has also been anonymized to remove any elements that could reveal project background. As such, the dataset does not pose copyright or privacy risks under the terms of licensing.

---

> > > > ### Comment · Reviewer_FDkf · 2025-08-08
> > > >
> > > > Thank you for the continued clarifications; I recommend that you include the number of unique drawings in FloorPlanCAD in your related work to avoid confusion.
> > > >
> > > > No further questions from me: I appreciate the additional qualitative and quantitative results, and would like to see them in the final paper.

---

### Official Review · Reviewer_WEoK · 2025-07-03

**Clarity:** 3
**Significance:** 3
**Originality:** 2
**Rating:** 5
**Confidence:** 5

**Summary:**

The work targets panoptic symbol spotting, a task useful in the context of engineering applications using real vector design data such as technical or architectural drawings. To address this task, the authors provide a dataset for training panoptic symbol spotting models that aims to close the gap w.r.t. the limits in data size in existing data collections for training deep learning models. The authors design and execute a new efficient data  collection and annotation pipeline to produce a 400K drawing-chunk dataset with symbol annotations across a variety of semantic classes and instances. The dataset is then employed to pre-train a high-performing panoptic symbol spotting model that shows favourable performance compared to a number of baselines.

**Questions:**

I would greatly appreciate the answers to the questions formulated in the "Weaknesses" part of the review. More specifically,
1. More details on the design choices regarding data collection and annotation, and the characteristics of the collected data need to be discussed. I do believe that the data is fair quality and useful, yet I'd like to paper to make a more compelling point about its qualities/properties.
2. A similar point could be made regarding the experimental evaluation, and in particular about data sampling, balancing, and overall preparation for this.

**Ethical Concerns:**

["NO or VERY MINOR ethics concerns only"]

**Final Justification:**

I have carefully read and understood the comments regarding my raised concerns, and have skimmed through the other reviews and discussions. I acknowledge that it was unfortunate that I could not answer the authors's concerns more promptly and engage in discussions.

I would like to raise my final rating to 5-accept, as overall my concerns were pretty much addressed.  Yet, below I'd like to outline my final concerns and recommendations over each of the questions. Perhaps the authors would  consider this worthy of further refinement for the final version of the paper.
**Q1, Q2:** I understand the idea, but as a reader I would like to have a clear idea what these "top-tier architectural design institutions" were, and which guidelines they follow for production of engineering drawings, organizing them into layers, etc, if such information is available. It would be useful to provide context wrt the data background and the type of practices involves in the original development of the drawings, otherwise the dataset risks _not representing global architectural diversity_  as outlined in the ethics review.

**Q3—Q5:** I think the answers are adequate, and as a reader, I would expect clear definitions in the paper.

**Q6—Q9:** I appreciate the authors making an effort to provide more experimental evidence wrt the performance of their models  and the statistics of the dataset. I thank the authors and hope this this information could be added to the final version of the paper.

**Limitations:**

Yes

**Paper Formatting Concerns:**

No concerns

**Quality:**

3

**Strengths And Weaknesses:**

Strengths:
1. The authors deliver the largest dataset for panoptic symbol spotting. Technically, the dataset could be subsequently used for other tasks such as vectorization of architectural (technical) drawings, due to its large size, real nature, and availability of vector ground-truth.
2. Several results around collection and benchmarking of the dataset are valuable. Specifically, the authors were able to collect the dataset quite efficiently, which has contributed to its large size. Moreover, the authors have demonstrated that using the dataset for pre-training for the panoptic symbol spotting task result in improved performance, even with "vanilla" models.

Weaknesses:
1. Insufficient description of the data selection and annotation procedure. Specifically,
— Where were the source (raw) data obtained from? Were these data paid for, and if yes, what are the financial costs associated with acquiring the data?
— In the main text, multiple references to "standardized drawings" were made, however, this term was not clarified and no reference was given to any definition from the literature.
— What do the authors identify as a layer? How specifically were layers selected/separated from the background drawing?
— What would be the implications of time savings on the quality of the final annotations? More generally, how trustworthy are the data annotations? I think the authors did not provide too much information regarding the inter-annotator agreement, or performance on any kind of a golden set, so it remains to be seen if the dataset has the claimed high quality.
— Is the taxonomy of semantic classes borrowed from FloorPlanCAD work or is it defined separately for this project?

2. Insufficient description of the design of the experimental evaluation:
— During the split of the dataset into the train/val/test parts, these three parts are ensured to contain drawings entirely, so as to prevent data leak from train to test, etc. However, this potentially limits the balancing quality across these sub-datasets, particularly in terms of instance/class numbers available for training, validation, or testing. Were the splits class-balanced?
— Because the CAD drawings offer symbols with varying geometric complexity, it might be interesting to study the impact of symbol complexity (e.g. number of primitives per symbol) on the ultimate performance. The same could be said about the rasterisation resolution of the images (e.g. how many pixels are sampled across a common symbol). Are there any results w.r.t. this?
— Are performance measures defined in the pixel space, or in the symbol space? E.g. a large (complex) symbol might significantly alter number as it is sampled with many pixels, yet it can only be a single number.

---

> ### Author Rebuttal · Authors · 2025-07-30
>
> **Q1: About the acquisition details of the dataset.**
>
> **A1:** Thank you for pointing this out. The raw CAD data was collected from engineering drawings provided by top-tier architectural design institutions. This ensures both the professional quality and diversity of the dataset, making it representative of real-world CAD design practices.
>
> The data was made available to us under a collaborative research agreement and no direct monetary cost was involved in acquiring the dataset.
>
> **Q2: About the concept of layer and the method used for layer extraction.**
>
> **A2:** In CAD drawings, a layer refers to a categorical division of graphical elements defined by engineers. The purpose of layer differentiation is to facilitate clearer organization of drawing content and enable selective rendering of specific components, thereby improving clarity and supporting more effective collaboration among team members.
>
> In our pipeline, we extract layer information by programmatically parsing the CAD files, leveraging metadata embedded in the DWG/DXF formats. These layers are then used to isolate structural components for downstream analysis.
>
> **Q3: About the definition and clarification of the term "standardized" drawings.**
>
> **A3:** We appreciate the reviewer's careful reading. The term "standardized" drawings refers to CAD drawings that follow a consistent layering convention, where each type of building component is assigned to a specific layer based on an internally defined standard. This standard ensures semantic consistency across drawings and facilitates automated parsing. A brief explanation of this standard is provided in Appendix A.2. We will refine this statement in the revised version.
>
> **Q4: About the reliability of annotations, including inter-annotator agreement and gold standard validation.**
>
> **A4:** We employed several strategies to ensure the reliability and accuracy of the annotations during the annotation process:
>
> (1) Annotation Guidelines: A comprehensive annotation manual was developed, detailing the definition and visual examples of each category. Annotators were trained using this guide to maintain consistency. A portion of the manual is included in Appendix A.1.
>
> (2) Quality Control: All annotations underwent a secondary verification step. Given the complexity of CAD drawings, we implemented a hierarchical review process, where a senior annotator re-checked each annotated file. This procedure helped to eliminate labeling inconsistencies and reduce human errors.
>
> (3) Customized Annotation Pipeline: As mentioned in paper 3.3, before manual labeling, we apply a pipeline that extracts inherent information such as layers and blocks to generate annotation suggestions. This avoids labeling from scratch, improves efficiency, and reduces human errors, especially in densely labeled CAD drawings.
>
> **Q5: About the semantic categories.**
>
> **A5:** The semantic categories used in our project were independently defined based on domain requirements. Our primary criterion was whether the categorized elements are essential for reconstructing a Building Information Model (BIM). While some categories align with prior work such as FloorPlanCAD, our classification scheme was tailored to industrial CAD drawings and encompasses additional component types specific to structural and architectural design.
>
> **Q6: About the train/val/test split of the dataset.**
>
> **A6:** Thank you for raising this important point. We split the dataset at the chunk level, rather than the drawing level, to form the train/val/test sets. This decision was made based on two key considerations:
>
> (1) Low risk of data leakage:  Chunks assigned to different splits do not spatially overlap, which theoretically prevents any form of data leakage between training and testing. Although some chunks may come from the same large drawing, their sparse details result in insignificant correlation between chunks, keeping the risk of data leakage low.
>
> (2) Improved balance across splits:  Variation in both class distribution and drawing area can cause imbalance when splitting at the drawing level. Chunk-level splitting helps ensure better balance in instance counts and spatial coverage across splits.
>
> As a result, class balance is easier to achieve and more controllable under the chunk-level splitting strategy.
>
> **Q7: About the relationship between symbol complexity and model performance.**
>
> **A7:** Thank you for raising this insightful point. We agree that understanding how symbol complexity and image resolution affect model performance is an interesting and helpful for understanding the problem.
>
> To investigate the impact of symbol complexity, we analyzed the number of primitives within each symbol as a proxy for complexity. Specifically, we partitioned the test symbols into eight subsets based on the percentile of primitive count and then computed evaluation metrics on each subset.
>
> | Symbol Complexity |  PQ  |  SQ  |  RQ  |
> | :------------------------: | :--: | :--: | :--: |
> |           \[1,4)           | 82.5 | 94.4 | 87.5 |
> |           \[4,5)           | 90.9 | 95.8 | 94.9 |
> |           \[5,6)           | 87.5 | 93.5 | 93.7 |
> |           \[6,9)           | 88.7 | 93.5 | 94.9 |
> |           \[9,13)          | 88.9 | 94.0 | 94.5 |
> |          \[13,21)          | 86.1 | 94.2 | 91.4 |
> |          \[21,46)          | 86.9 | 93.2 | 93.3 |
> |           \[46,)           | 77.1 | 85.3 | 90.4 |
>
> From the results, we can find that there exists a certain correlation between symbolic complexity and model performance. In brief, both extremely simple and extremely complex symbols represent performance bottlenecks for the model.
>
> For symbols with low complexity within the range [1, 4), the model exhibits relatively low recognition quality (RQ), likely due to the lack of distinctive structural features. Within the moderate complexity range [4, 46), the model demonstrates relatively stable performance. However, in the highest complexity split [46, ), the performance degradation is primarily attributed to a decline in segmentation quality (SQ), as the high structural complexity leads to intricate and ambiguous symbol boundaries in the symbolic space.
>
> **Q8: About the relationship between image resolution and model performance.**
>
> **A8:** Thank you for raising this insightful point. We conducted a set of experiments to evaluate the influence of raster image resolution on model performance. Specifically, we varied the resolution used for rasterizing the input vector graphics while keeping all other components of the pipeline unchanged. The results are summarized as follows.
>
> | Raster Size |  PQ  |  SQ  |  RQ  |
> | :------------------: | :--: | :--: | :--: |
> |        384×384       | 69.6 | 79.0 | 88.1 |
> |        700×700       | 86.2 | 93.0 | 92.6 |
> |       1024×1024      | 85.8 | 92.3 | 92.9 |
>
> The results suggest that increasing the resolution is beneficial to model performance when starting from a relatively low baseline. However, beyond a certain threshold further increasing the resolution yields diminishing returns and may lead to degradation in efficiency.
>
> This phenomenon can be attributed to the inherent characteristics of CAD drawings: since they consist primarily of discrete, line-based primitives, the amount of spatial detail is inherently limited. Once the resolution is sufficient to capture and distinguish these primitives, further increases do not yield additional useful information.
>
> **Q9: The evaluation metrics are defined in pixel space or symbolic space?**
>
> **A9:** Thank you for raising this question. The evaluation metrics for panoptic symbol spotting are defined in the symbolic space. Their formal definitions can be found in [1], and they are commonly adopted as standard practice in prior work in the panoptic symbol spotting task.
> More specifically, the key distinction between evaluation in pixel space and symbolic space lies in the way IoU is computed. For a symbol $s$ represented as a set of primitives $s= \lbrace e_1,e_2,… \rbrace $, the IoU between a predicted symbol $s_p$ and a ground truth symbol $s_g$ is calculated as:
>
> $$
> \mathrm{IoU}(s_p, s_g) = \frac{\sum_{e_i \in s_p \cap s_g} \log{(1+L(e_i))}}{\sum_{e_i \in s_p \cup s_g} \log{(1+L(e_i))}}
> $$
>
> where $L(e_i)$ denotes the length of the primitive $e_i$.
>
> This formulation ensures that the metric is sensitive to geometric correctness at the symbol level, while also addressing the issue of symbol size imbalance. The use of a logarithmic scaling function $\log{(1+L(e_i))}$ prevents large symbols from disproportionately dominating the metric.
>
> **References**
>
> [1] Zhiwen Fan, Lingjie Zhu, Honghua Li, Xiaohao Chen, Siyu Zhu, and Ping Tan. Floorplancad: A large-scale cad drawing dataset for panoptic symbol spotting. In Proceedings of the IEEE/CVF international conference on computer vision, pages 10128–10137, 2021.

---

### Comment · Area_Chair_ADcc · 2025-08-04

Dear Reviewers, the authors have provided a detailed rebuttal including a number of experimental results. Please read through it thoroughly and discuss potentially open points. Does the rebuttal change you assessment of the paper?
Best, AC

---

### Decision · Program_Chairs · 2025-09-17

**Decision:**

Accept (poster)

**Comment:**

The manuscript addresses the task of panoptic symbol detection, as for example needed for technical or architectural drawings. To address this task, the paper introduces ArchCAD-400K, a large-scale dataset of architectural CAD files from the internet, which can be used to train foundation models. Therefore, the paper provides an efficient data collection and annotation pipeline. The dataset has symbol annotations across a variety of semantic classes and instances. The paper further provides a model pre-trained on this dataset and evaluated on panoptic symbol spotting.
While seeing the merits of the dataset and model, the reviewers were initially critical about the paper, mostly because of limited description of the dataset details, collection process and evaluation. However, these concerns could be resolved during the discussion phase, leaving the paper with positive scores after the rebuttal and discussion.